# RANKCLIP: RANKING-CONSISTENT LANGUAGE-IMAGE PRETRAINING

## ABSTRACT

Self-supervised contrastive learning models, such as CLIP, have set new benchmarks for vision-language models in many downstream tasks. However, their dependency on rigid one-to-one mappings overlooks the complex and often multifaceted relationships between and within texts and images. To this end, we introduce **RANKCLIP**, a novel pretraining method that extends beyond the rigid one-to-one matching framework of CLIP and its variants. By extending the traditional pair-wise loss to list-wise, and leveraging both in-modal and cross-modal ranking consistency, RANKCLIP improves the alignment process, enabling it to capture the nuanced many-to-many relationships between and within each modality. Through comprehensive experiments, we demonstrate the effectiveness of RANKCLIP in various downstream tasks, notably achieving significant gains in zero-shot classifications over state-of-the-art methods, underscoring the importance of this enhanced learning process.

## 1 INTRODUCTION

In the realm of computer vision (CV) (Voulodimos et al., 2018), natural language processing (NLP) (Chowdhary & Chowdhary, 2020), and multimodal deep learning (Jabeen et al., 2023; Zhao et al., 2023; Chen et al., 2024a), the alignment between visual and textual modalities (Singh et al., 2022; Chen et al., 2024b) has emerged as a cornerstone for downstream applications, ranging from image captioning (Ghandi et al., 2023) to zero-shot classification (Pourpanah et al., 2022). Contrastive Language-Image Pretraining (CLIP) (Radford et al., 2021) marks a significant advancement in this field, demonstrating incredible performance from training on large amounts of text-image pairs to create self-supervised models that understand (Hendrycks et al., 2021a;b; Chen et al., 2024c) and generate (Ramesh et al., 2021; Crowson et al., 2022) descriptions of visual contents. Following the success of this contrastive learning paradigm, many recent works have been developed upon the original CLIP. More specifically, these enhancements focus on optimizing data efficiency through intrinsic supervision (Li et al., 2021), as well as improving downstream performance via cross-modal late interaction (Yao et al., 2021), hierarchical feature alignment (Gao et al., 2022), geometric consistency regularization (Goel et al., 2022), additional learning (Mu et al., 2022), adaptive loss (Yang et al., 2023), hierarchy-aware attentions (Geng et al., 2023), and softer cross-modal alignment (Gao et al., 2024).

Despite the improvements, these methods often have reliance on strict *pairwise, cross-modal, and one-to-one* mappings between images and texts, overlooking the actual *many-to-many* relationships that exist both *cross-modal* and *in-modal* in real-world data (Chun, 2023). For example, as shown in Fig. 1, while pretrained models like CLIP can correctly classify `dog`, `cat` and `airplane`, they do not necessarily learn that `dog` and `cat` are more close to each other than `dog` and `airplane`, in terms of both in-modal (`dog` text is more similar to `cat` text than to `airplane` text) and cross-modal (`dog` text is more matched to `cat` image than to `airplane` image) similarities. Because it is rooted from the current contrastive loss that only the correct pairs are optimized while the rest of the unmatched pairs are treated the same, resulting in a large amount of information not used and unknown to the model during and after the training process.

Recognizing the complex *many-to-many* relationships as well as the rich information contained within both *in-modal* and *cross-modal* data, we introduce **Rank**ing-**C**onsistent **L**anguage-**I**mage **P**retraining, (**RANKCLIP**), which employs *ranking consistency* to learn and optimize similarity levels both between (cross-modal) and within (in-modal) the text-image pairs.

The concept of ranking consistency stems from the simple observations that similar texts often correlate with similar images, as seen with the `dog`, `cat` and `airplane` example in Fig. 1. It effectively captures secondary similarity relationships among unmatched pairs, enabling the model to learn *more efficiently for free* compared to relying solely on matched pairs. Ranking consistency is conveniently modeled as an additional loss term to the traditional contrastive loss, requiring no extra external modules. It acts as a plug-and-play improvement for many existing methods, including those focusing on data-efficiency (Li et al., 2021), potentially boosting performance in both efficiency and effectiveness.

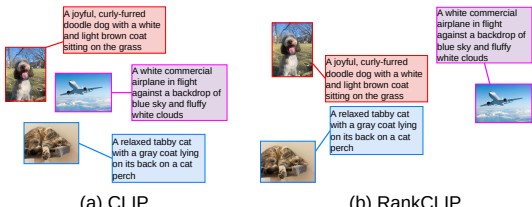

(a) CLIP          (b) RankCLIP

Figure 1: Comparison of learning outcomes between (a) CLIP and (b) RANKCLIP using three text-image pairs: `dog` (red), `cat` (blue), and `airplane` (magenta), where matched pairs share the same color boundaries. CLIP treats all unmatched relationships equally, failing to distinguish similarities better between dog and cat versus airplane. RANKCLIP addresses this by training with ranking consistency, enhancing its understanding of complex relationships.

The main contributions of this paper are: 1) RANKCLIP, a novel contrastive pretraining method that uses ranking consistency to exploit the many-to-many relationships within data, thereby enhancing performance in downstream tasks such as zero-shot classification and retrieval accuracy; and 2) through comprehensive experiments conducted on multiple datasets, we demonstrate the superior effectiveness of RANKCLIP in improving pretraining model performance without requiring any additional data or computational resources.

## 2   RELATED WORK

Vision-language pretraining has witnessed significant advancements over the past years (Chen et al., 2023; Du et al., 2022; Long et al., 2022). Models such as CLIP (Radford et al., 2021), ALIGN (Jia et al., 2021) and FLAVA (Singh et al., 2022) have pioneered the contrastive learning paradigm applied with text-image pairs, showcasing remarkable performance and robustness in downstream tasks. Many follow-up works, mostly built upon CLIP, have been proposed since then. Li et al. (2021) introduced DeCLIP, improving zero-shot performance through intrinsic supervision. FILIP (Yao et al., 2021) advances CLIP's alignment between image patches and text with a cross-modal interaction mechanism. Gao et al. (2022) developed PyramidCLIP, using hierarchical feature alignment to boost model efficiency and performance. Additionally, SLIP (Mu et al., 2022) merges self-supervised learning with CLIP pre-training for improved visual representation and accuracy. Goel et al. (2022) introduced CyCLIP, augmenting CLIP with geometric consistency regularizers to enhance robustness and performance under varied conditions.

Recently, Yang et al. (2023) introduced ALIP, an adaptive pre-training model that enhances language-image alignment using raw text and synthetic captions with dynamic adjustments. Hi-CLIP (Geng et al., 2023) refines CLIP by adding hierarchy-aware attentions to uncover semantic hierarchies in images and texts. EqSim (Wang et al., 2023) incorporates equivariance loss into vision-language models, significantly improving sensitivity to semantic changes in image-text pairs. Additionally, SoftCLIP (Gao et al., 2024) softens CLIP's one-to-one constraint, enabling more flexible cross-modal alignment through fine-grained adjustments.

Compared with existing approaches, RANKCLIP sets itself apart by fully leveraging the *many-to-many* relationships within each batch of text-image pairs, promoting learning from both matched and unmatched pairs with varying similarities by integrating in-modal and cross-modal *list-wise ranking consistencies* into the contrastive training objective. Crucially, RANKCLIP diverges from existing models' pair-wise training objective by adopting a global, list-wise optimization approach. In other words, it considers the rankings of all images and texts collectively within each batch, rather than focusing on pairwise similarities as seen in other methods.

## 3   CLIP PRELIMINARIES

CLIP (Radford et al., 2021) has been a prominent method for learning detailed multimodal representations through the alignment of images and texts. Given a set $\mathcal{D} = \{(V_j, T_j)\}_{j=1}^{N}$ of $N$ image-text

pairs, where $V_j$ denotes an image and $T_j$ is the corresponding text, the goal is to learn representations that map semantically similar images and texts closer in the embedding space, while dissimilar pairs are distanced apart. More specifically, the foundational CLIP model employs two encoders: an image encoder $f_I : \mathcal{I} \to \mathbb{R}^m$ that processes raw images into visual embeddings and a text encoder $f_T : \mathcal{T} \to \mathbb{R}^n$ which encodes textual data into text embeddings. Then both the text and visual features are projected to a latent space with identical dimension. Formally, the embeddings for a text-image pair $(V_j, T_j)$ are denoted as $v_k = f_I(V_j)$ and $t_j = f_T(T_j)$, respectively. The embeddings are then normalized to lie on an unit hypersphere by enforcing $l_2$-norm constraint:

$$\hat{v}_j = \frac{v_j}{\|v_j\|_2}, \quad \hat{t}_j = \frac{t_j}{\|t_j\|_2}. \tag{1}$$

so that the magnitude information is erased and only direction is preserved.

To align the image and text representations, a contrastive loss function, typically a variant of the InfoNCE loss (Oord et al., 2018), which optimizes the similarity of the matched pair against unmatched pairs, is utilized, i.e.:

$$\mathcal{L}_{\text{CLIP}} = -\frac{1}{2N} \sum_{j=1}^{N} \left[ \log \underbrace{\frac{\exp(\hat{v}_j^\top \hat{t}_j / \tau)}{\sum_{k=1}^{N} \exp(\hat{v}_j^\top \hat{t}_k / \tau)}}_{\textcircled{1}} + \log \underbrace{\frac{\exp(\hat{t}_j^\top \hat{v}_j / \tau)}{\sum_{k=1}^{N} \exp(\hat{t}_j^\top \hat{v}_k / \tau)}}_{\textcircled{2}} \right] \tag{2}$$

where the first term $\textcircled{1}$ contrasts images with the texts, the second term $\textcircled{2}$ contrasts texts with the images, and $\tau$ denotes a temperature scaling parameter that adjusts the concentration of the distribution. The optimization of Eqn. (2) results in embeddings where the cosine similarity between matched image-text pairs is maximized in comparison to unmatched pairs, thus achieving the desired alignment in the joint embedding space.

Despite the efficacy of CLIP in learning correlated multimodal embeddings, it inherently relies on strict pairwise matched comparisons and fails to capture the more complex, fine-grained nature of semantic similarity within and across modalities that are generally treated as unmatched. This observation motivates the development of RANKCLIP, which innovates beyond binary pairwise contrasts to consider holistic listwise consistency within and across modalities.

## 4 RANKCLIP

RANKCLIP efficiently leverages the many-to-many relationships in real-world data by focusing on both matched and unmatched pairs. As shown in Fig. 2, it not only identifies if an image-text pair matches but also assesses their relative semantic similarities to other images and texts of both modalities in the dataset through self-supervised ranking consistency. Uniquely, RANKCLIP employs a list-wise loss for training batches, distinguishing it from other methods that solely rely on pair-wise relationships, as discussed in §2.

### 4.1 RANKING MODEL FORMULATION

RANKCLIP leverages the Plackett-Luce (PL) ranking model Plackett (1975); Luce (2005); Guiver & Snelson (2009) to estimate the probability distribution over rankings for every image-text pair $(V_i, T_j)$, so that the consistency in their relative ordering with respect to a reference ranking can be measured. Specifically, for a given data pair, whether it is in-modal (image-image, text-text), or cross-modal (image-text), we calculate its in- or cross-modal cosine similarity $S_{ij}$ to serve as the score when measuring the alignment of its ranking with respect to another reference ranking $y_{\text{ref}}$.

Following Plackett (1975), we first sort the reference ranking in a descending order to construct the optimal ranking $y^*$, and assume that the ego ranking $y$ is sampled from $y^*$. Thus the probability that item $d$ with score $S_{ij}$ is ranked $k^{\text{th}}$ in the ego ranking $y$ from a set of items $\mathcal{D}$ is the score of $e^{S_{ij}}$ divided by the sum of scores for the items that have not been placed yet:

$$\pi(d \mid y_{1:k-1}, y_{\text{ref}}, \mathcal{D}) = \frac{e^{S_{ij}}}{\sum_{d' \in \mathcal{D} \setminus y_{1:k-1}} e^{S'_{ij}}}, \tag{3}$$

where $y_{1:k-1} = [y_1, y_2, ..., y_{k-1}]$ denotes the set of items ranked before $d$. In addition, we propose a decaying factor $\mu = 1/\log(k+1)$ to scale the loss, so that the top-ranked items can obtain

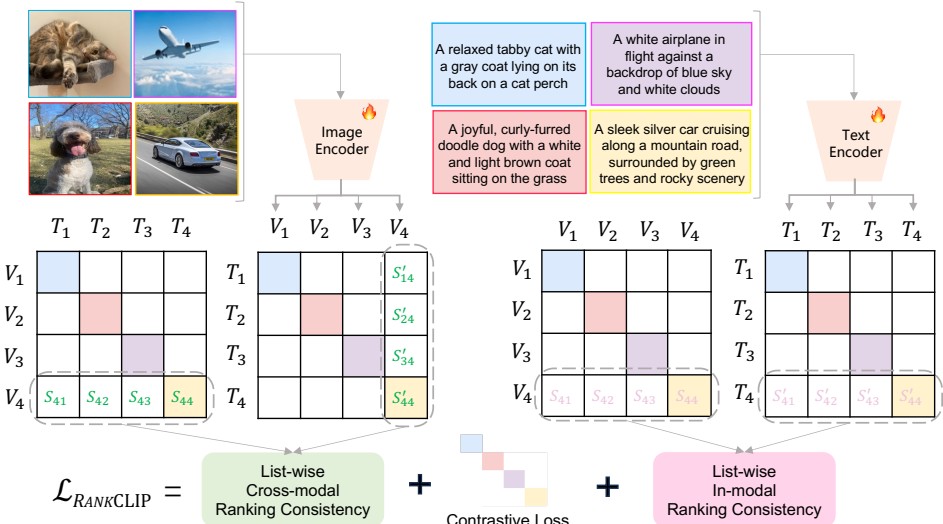

Figure 2: Illustrative overview of RANKCLIP. Unlike conventional contrastive loss, which includes only the middle term, RANKCLIP introduces both cross-modal and in-modal consistency terms by minimizing a self-supervised, list-wise ranking loss. Paired images and texts are indicated by matching contour line colors. $V$, $T$, and $S$ represent image embeddings, text embeddings, and similarity scores, respectively.

higher weights: Consequently, the probability of the entire ranking $y$ is the product of all individual placement probabilities:

$$\mathcal{P}(y, y_{\text{ref}}) = \prod_{k=1}^{K} \mu \cdot \pi(y_k \mid y_{1:k-1}, \mathbf{y}_{\text{ref}}, \mathcal{D}). \tag{4}$$

RANKCLIP's objective is to maximize the consistency log-likelihood of the list ranking in one modality towards the reference ranking (from the same/in-modal and different/cross-modal data), which conveniently aligns with minimizing the negative log-likelihood loss:

$$\mathcal{L}_{\text{PL}} = -\log \mathcal{P}(y, y_{\text{ref}}) \tag{5}$$

## 4.2 CROSS-MODAL CONSISTENCY RANKING

As illustrated by the green box in Fig. 2, RANKCLIP utilizes secondary relationships between unmatched visual and textual representations by constructing a list-wise rank loss. This approach ensures that the semantic similarity rankings between one image and multiple texts align with those between one corresponding text and multiple images. For example, as shown in Fig. 1, from the dog perspective, the semantic distance between dog image and cat text is closer compared to the plane text. This relationship should also apply between the dog text and the cat, plane images. Mathematically, Eq. (5) can be specified as:

$$\mathcal{L}_{\text{cross-modal}} = -\log \mathcal{P}(\mathbf{y}_{\text{image-text}}, \mathbf{y}_{\text{text-image}}) \tag{6}$$
$$= -\log \mathcal{P}(\hat{\mathbf{v}} \cdot \hat{\mathbf{t}}^{\mathbf{T}}, \hat{\mathbf{t}} \cdot \hat{\mathbf{v}}^{\mathbf{T}}) \tag{7}$$

By optimizing Eq. (6), RANKCLIP enhances its ability to bridge the semantic gap between modalities by leveraging nuanced unmatched correlations. This can also be viewed as learning a symmetric cosine-similarity matrix, further reinforcing semantic consistency across modalities.

## 4.3 IN-MODAL CONSISTENCY RANKING

The pink box in Fig. 2 highlights the in-modal consistency component of the proposed rank loss. RANKCLIP ensures semantic consistency within each modality – image to image and text to text – enhancing the use of secondary unmatched relationships as an optimization objective. The underlying principle is that similar images should correspond to similar texts. For example, in Fig. 1, from the dog image perspective, the cat image is the most similar, followed by the plane image. This

relationship should hold true for their corresponding texts as well, where we utilize this to construct our $y$ and $y_{\text{ref}}$ from Eq. (5). Mathematically, Eq. (5) can be specified as:

$$\mathcal{L}_{\text{in-modal}} = -\log \mathcal{P}(\mathbf{y}_{\text{text-text}}, \mathbf{y}_{\text{image-image}}) \tag{8}$$

$$= -\log \mathcal{P}(\hat{\mathbf{t}} \cdot \hat{\mathbf{t}}^{\mathbf{T}}, \hat{\mathbf{v}} \cdot \hat{\mathbf{v}}^{\mathbf{T}}) \tag{9}$$

where $\hat{\mathbf{t}}$ and $\hat{\mathbf{v}}$ are the text and image batch embedding matrix, respectively. Via Eq. (8), the model can efficiently leverage the nuanced in-modal relationships to learn a richer and more structured semantic representation.

## 4.4 RANKCLIP LOSS

Combining both cross-modal and in-modal consistency with the traditional contrastive loss (more details in Appendix 3), the complete rank loss is thus formulated as:

$$\mathcal{L}_{\text{RANKCLIP}} = \mathcal{L}_{\text{CLIP}} + \lambda_1 \mathcal{L}_{\text{in-modal}} + \lambda_2 \mathcal{L}_{\text{cross-modal}} \tag{10}$$

which is also depicted in Fig. 2. By supplementing the pairwise contrastive loss with cross-modal and in-modality ranking consistency loss, RANKCLIP systematically organizes embeddings to fully exploit both global and fine-grained secondary unmatched relationships, which enhances the learning of more informative and accurate representations, better supporting downstream multi-modal tasks. The complete RANKCLIP is detailed in Algorithm 1.

---

**Algorithm 1** Pseudo-code of RANKCLIP loss in a Python-like style.

---

```
# emb_pred: predictions from the model, shape [embs_length, embs_length]
# emb_true: ground truth labels, shape [embs_length, embs_length]

def rank_loss(emb_pred, emb_true):
    # Shuffle for randomised tie resolution
    emb_pred_shuff = emb_pred[:, random_indices]
    emb_true_shuff = emb_true[:, random_indices]
    # Record the rank label index
    emb_true_sorted, indices = emb_true_shuff.sort(descending=True, dim
        =-1)
    # Ranking the pred embedding by the true indices
    preds_sorted = gather(emb_pred_shuff, dim=1, index=indices)
    # Implementation of the Eq.1, Eq.2 and Eq.3
    max_pred_values, _ = preds_sorted.max(dim=1, keepdim=True)
    preds_sorted_minus_max = preds_sorted - max_pred_values
    cumsums = cumsum(preds_sorted_minus_max.exp().flip(dims=[1]), dim=1).
        flip(dims=[1])
    loss = (log(cumsums) - preds_sorted_minus_max) * scale_factor
    return mean(sum(loss, dim=1))

# Cross-modal embeddings
logits_text_per_image=image_embeds @ text_embeds.T
logits_iamge_per_text=logits_text_per_image.T
# In-modal embeddings
logits_image_per_image=image_embeds @ image_embeds.T
logits_text_per_text=text_embeds @ text_embeds.T
# Compute the cross-modal rank loss
Cross_modal_loss=rank_loss(logits_text_per_image,logits_image_per_text)+
    rank_loss(logits_image_per_text, logits_text_per_image)
# Compute the in-modal rank loss
In_modal_loss=rank_loss(logits_image_per_image,logits_text_per_text)+
    rank_loss(logits_text_per_text, logits_image_per_image)
# Rank loss
Rank_loss=Contrastive_loss+Cross_modal_loss+In_modal_loss
```

---

Table 1: Zero-shot top-1, top-3 and top-5 classification accuracy on CIFAR-10, CIFAR-100 and ImageNet1K. Relative to CLIP, RANKCLIP achieves higher accuracy with *average* top-1, top-3, and top-5 improvements of +2.46%, +2.25%, and +2.40%, respectively. RANKCLIP also outperforms ALIP consistently across the datasets.

| | CIFAR-10 | | | CIFAR-100 | | | ImageNet1K | | |
|---|---|---|---|---|---|---|---|---|---|
| | Top1 | Top3 | Top5 | Top1 | Top3 | Top5 | Top1 | Top3 | Top5 |
| CLIP | 36.35% | 70.28% | 85.02% | 12.22% | 24.93% | 33.56% | 12.08% | 21.86% | 27.48% |
| ALIP | 35.71% | **72.39%** | **88.77%** | 13.67% | 27.10% | 34.76% | 15.62% | 26.90% | 32.50% |
| RANKCLIP | **37.03%** | 67.67% | 83.09% | **13.98%** | **27.70%** | **36.17%** | **17.02%** | **28.44%** | **33.99%** |
| | (+0.68%) | (-2.61%) | (-1.93%) | (+1.76%) | (+2.77%) | (+2.61%) | (+4.94%) | (+6.58%) | (+6.51%) |

Table 2: Zero-shot top-1, top-3 and top-5 classification accuracy on variants of ImageNet1K that have *natural distribution shifts*. Relative to CLIP, RANKCLIP achieves higher accuracy with *average* top-1, top-3, and top-5 improvements of +3.15%, +4.19%, and +4.66%, respectively. Notice that the average improvements are more significant than when tested on ImageNet1K without distribution shift, indicating better robustness.

| | ImageNetV2 | | | ImageNetSketch | | | ImageNet-A | | | ImageNet-R | | |
|---|---|---|---|---|---|---|---|---|---|---|---|---|
| | Top1 | Top3 | Top5 | Top1 | Top3 | Top5 | Top1 | Top3 | Top5 | Top1 | Top3 | Top5 |
| CLIP | 12.11% | 22.66% | 28.57% | 3.20% | 7.00% | 9.83% | 3.16% | 8.81% | 13.04% | 11.34% | 21.38% | 27.10% |
| ALIP | 15.62% | 27.34% | 32.82% | 5.10% | 10.37% | 14.01% | 3.53% | 9.14% | 13.61% | 14.25% | 25.74% | 32.43% |
| RANKCLIP | **17.03%** | **28.60%** | **34.18%** | **5.82%** | **11.35%** | **14.87%** | **3.82%** | **9.16%** | **13.77%** | **15.74%** | **27.51%** | **34.36%** |
| | (+4.92%) | (+5.94%) | (+5.61%) | (+2.62%) | (+4.35%) | (+5.04%) | (+0.66%) | (+0.35%) | (+0.73%) | (+4.40%) | (+6.13%) | (+7.26%) |

# 5 EXPERIMENTS

## 5.1 EXPERIMENTAL SETUP

**Baselines.** The most direct baseline to RANKCLIP is the original CLIP (Radford et al., 2021). In addition, to further demonstrate the superior performance of RANKCLIP, we include ALIP (Yang et al., 2023), which leverages synthetic captions to enhance vision-language representation learning. More specifically, it employs a unique architecture that dynamically adjusts sample and pair weights to mitigate the impact of noisy or irrelevant data, which is quite orthogonal to our approach. The training procedures and parameters of all models are detailed in Appendix A.

**Pretraining dataset.** Both baseline models, CLIP (Radford et al., 2021), ALIP (Yang et al., 2023) and the proposed RANKCLIP are pretrained on the Conceptual Captions 3M (CC3M) dataset (Sharma et al., 2018), which contains around 3.3 million text-image pairs. Despite being much smaller than CLIP's initial dataset (Ilharco et al., 2021), CC3M effectively supports the development of pretrained models with strong zero-shot capabilities and is widely used in existing language-image pretraining research (Carlini & Terzis, 2021; Li et al., 2021; Tejankar et al., 2021; Mu et al., 2022; Goel et al., 2022). Additionally, as discussed later in §6.2, we trained both CLIP and RANKCLIP on 15 million text-image pairs, filtered from YFCC100M (Thomee et al., 2016), referred to as YFCC15M, to conduct an ablation study on the impact of data size.

## 5.2 ZERO-SHOT CLASSIFICATION

Zero-shot capability is one of the most significant improvements that CLIP achieves. Thus in this section, we first evaluate the zero-shot classification performance of CLIP, ALIP and the proposed RANKCLIP. Following (Goel et al., 2022), we conduct our experiments on CIFAR-10 (Krizhevsky et al., 2009), CIFAR-100 (Krizhevsky et al., 2009), and ImageNet1K (Deng et al., 2009; Russakovsky et al., 2015) dataset.

As shown in Table 1, RANKCLIP consistently outperforms CLIP across CIFAR-10, CIFAR-100, and ImageNet1K datasets. Relative to CLIP, RANKCLIP shows average improvements of +3.15%, +4.19%, and +4.66% in top-1, top-3 and top-5 metrics, respectively. Particularly on the more challenging ImageNet1K dataset, RANKCLIP improves relative top-1 accuracy by +4.94% over CLIP, highlighting the effectiveness of the proposed ranking consistency in enhancing language-image alignment and understanding with the same amount of training data. The two cases where RANKCLIP does not excel are the top-3 and top-5 accuracy on CIFAR-10. However, this is likely

Table 3: Linear probing top-1 accuracy on 11 downstream datasets. RANKCLIP achieves higher accuracy than CLIP with an average improvement of +1.30%. RANKCLIP also outperforms ALIP, although less significantly.

| | CIFAR-10 | CIFAR-100 | DTD | FGVGAircraft | Food101 | GTSRB | ImageNet1K | OxfordPets | SST2 | STL10 | SVHN | Average |
|---|---|---|---|---|---|---|---|---|---|---|---|---|
| CLIP | 72.40% | 48.43% | 49.89% | 26.10% | 48.59% | 65.20% | 77.49% | 49.74% | 53.71% | 83.59% | 44.80% | 56.37% |
| ALIP | 73.87% | 51.00% | 58.09% | 27.72% | 49.74% | 60.34% | 73.14% | 59.36% | 53.98% | 87.94% | 38.07% | 57.56% |
| RANKCLIP | 72.54% | 49.16% | 53.24% | 24.99% | 47.11% | 63.37% | 86.40% | 54.10% | 54.09% | 86.10% | 43.30% | **57.67%** |
| | (+0.14%) | (+0.73%) | (+3.35%) | (-1.11%) | (-1.48%) | (-1.83%) | (+8.91%) | (+4.36%) | (+0.38%) | (+2.51%) | (-1.50%) | (+1.30%) |

Table 4: Zero-shot image and text retrievals on Flickr30K and MSCOCO. RANKCLIP achieves higher accuracy than both CLIP and ALIP on most cases.

| | Flickr30K | | | | | | MSCOCO | | | | | |
|---|---|---|---|---|---|---|---|---|---|---|---|---|
| | Text Retrieval | | | Image Retrieval | | | Text Retrieval | | | Image Retrieval | | |
| | R@1 | R@5 | R@10 | R@1 | R@5 | R@10 | R@1 | R@5 | R@10 | R@1 | R5 | R@10 |
| CLIP | 84.00% | 88.70% | 91.00% | 8.70% | 16.90% | 21.20% | 82.06% | 85.24% | 87.82% | 5.04% | 12.98% | 18.32% |
| ALIP | 84.40% | 90.00% | 92.50% | 9.40% | 17.60% | 21.30% | 82.56% | 86.04% | 88.26% | 6.08% | 13.96% | 19.38% |
| RANKCLIP | 84.10% | 89.40% | 91.90% | 8.10% | 16.40% | 21.70% | 82.90% | 85.68% | 88.00% | 5.60% | 13.20% | 18.02% |
| | (+0.10%) | (+0.70%) | (+0.90%) | (-0.60%) | (-0.50%) | (+0.50%) | (+0.84%) | (+0.44%) | (+0.18%) | (+0.56%) | (+0.22%) | (-0.30%) |

because CIFAR-10 with top-3 and top-5 metrics is much simpler, reducing the demand for a deeper model understanding.

Additionally, we observe that RANKCLIP consistently outperforms ALIP, suggesting that our ranking consistency more effectively enhances text-image representations and alignments compared to the synthetic captions proposed in ALIP. Another trend we observe is that RANKCLIP shows the most significant improvement in top-1 accuracy compared to top-3 and top-5. Considering the real-world emphasis on the topmost model output, RANKCLIP is likely to offer considerable advantages in practical applications.

## 5.3 ROBUSTNESS TO DISTRIBUTION SHIFTS

To evaluate the robustness of RANKCLIP under distribution shifts, we test it alongside CLIP and ALIP across four ImageNet variants, including ImageNetV2 (Recht et al., 2019), ImageNetSketch (Wang et al., 2019), ImageNet-A (Hendrycks et al., 2021b), and ImageNet-R (Hendrycks et al., 2021a), which are designed to assess resilience to different distribution shifts.

As shown in Table 2, RANKCLIP outperforms both CLIP and ALIP consistently. Notably, relative to CLIP, RANKCLIP's accuracy improvements in shifted conditions are +3.15% (top-1), +4.19% (top-3), and +4.66% (top-5), surpassing its performance in standard settings (Table 1) of +2.46% (top-1), +2.25% (top-3), and +2.40% (top-5), indicating the even more superior performance in robustness under distribution shifts.

## 5.4 LINEAR PROBING

We also evaluate whether the introduced ranking consistency retains its advantages when supplemented with additional in-domain supervision. Specifically, we use linear probing, where the pretrained encoders from CLIP, ALIP, and RANKCLIP remain unchanged while a logistic regression classifier is trained on domain-specific datasets. We evaluate on a suite of 11 standard image classification datasets as our in-domain datasets, which include CIFAR-10, CIFAR-100, Describable Textures Dataset (DTD) (Cimpoi et al., 2014), Fine-Grained Visual Classification of Aircraft (FGVG-Aircraft) (Maji et al., 2013), Food101 (Bossard et al., 2014), German Traffic Sign Detection Benchmark (GTSDB) (Stallkamp et al., 2012), ImageNet1K (Deng et al., 2009; Russakovsky et al., 2015), OxfordPets (Parkhi et al., 2012), Stanford Sentiment Treebank v2 (SST2) (Socher et al., 2013), STL-10 (Coates et al., 2011), and Street View House Numbers (SVHN) (Netzer et al., 2011) dataset.

Table 3 indicates that RANKCLIP consistently outperforms CLIP, with relative improvements ranging from +0.14% to +8.91% and an average accuracy increase of +1.30%. When compared to ALIP, RANKCLIP also shows better performance on average, though the gains are relatively modest.

Table 5: Ablation zero-shot classification accuracy of cross-modal-only model RANKCLIP$_C$ and in-modal-only model RANKCLIP$_I$ on CIFAR-10, CIFAR-100 and ImageNet1K datasets. Bold indicates the best performance, while blue indicates the second best.

| | CIFAR-10 | | | CIFAR-100 | | | ImageNet1K | | |
|---|---|---|---|---|---|---|---|---|---|
| | Top1 | Top3 | Top5 | Top1 | Top3 | Top5 | Top1 | Top3 | Top5 |
| CLIP | 36.35% | **70.28%** | **85.02%** | 12.22% | 24.93% | 33.56% | 12.08% | 21.86% | 27.48% |
| RANKCLIP | 37.03% | 67.67% | 83.09% | **13.98%** | **27.70%** | **36.17%** | **17.02%** | **28.44%** | **33.99%** |
| RANKCLIP$_I$ | **37.47%** | 69.89% | 84.53% | 13.89% | 27.34% | 35.90% | 16.66% | 27.63% | 33.15% |
| RANKCLIP$_C$ | 28.26% | 59.65% | 75.45% | 13.29% | 26.85% | 34.71% | 16.98% | 28.25% | 33.90% |

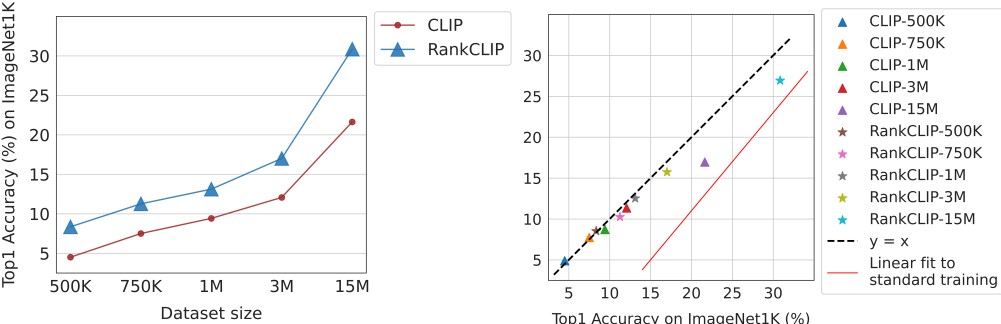

Figure 3: Ablation studies of CLIP and RANKCLIP trained with different data sizes. *Left*: zero-shot top-1 classification accuracy on ImageNet1K with various data sizes randomly sampled from CC3M. RANKCLIP consistently outperforms CLIP with significant margins. *Right*: zero-shot top-1 classification accuracy on ImageNet1K (horizontal axis) and ImageNet1K-R (vertical axis). RANKCLIP demonstrates better robustness as well as accuracy.

## 5.5 ZERO-SHOT IMAGE-TEXT RETRIEVAL

In the final part of our experiments, we assess RANKCLIP on zero-shot cross-modal retrieval tasks (image-to-text and text-to-image) using the Flickr30k (Plummer et al., 2015) and MSCOCO (Lin et al., 2014) datasets. As shown in Table 4, RANKCLIP generally outperforms the two baseline methods, though improvements are less significant compared to earlier results in Table 1, Table 2 and Table 3. The relatively modest gains in retrieval tasks may stem from the complex requirements of discerning image-text similarities across varying resolutions and object details, a significant departure from the simpler demands of image classification tasks. Despite this, the overall improvement highlights RANKCLIP's advantage, thanks to the deeper insights provided by ranking consistency in the language-image training process.

## 6 ABLATION STUDIES

### 6.1 ABLATION ON LOSS COMPONENTS

To further assess the effectiveness of the proposed ranking consistency, we developed two variants of RANKCLIP: RANKCLIP$_C$, focusing solely on cross-modal consistency with $\lambda_i = 0$, and RANKCLIP$_I$, emphasizing in-modal consistency with $\lambda_c = 0$. Both models underwent the same pretraining as outlined in Appendix A and were tested in a zero-shot classification experiment on ImageNet1K as in §5.2. The results are shown in Table 5, with bold font indicating the best performance, and blue color representing the second best results. We can see that, while RANKCLIP achieves the best performance, both RANKCLIP$_C$ and RANKCLIP$_I$ demonstrate notable improvements over CLIP. Interestingly, RANKCLIP$_I$ matches the performance of RANKCLIP$_C$, highlighting the often-underestimated value of in-modal consistency in enhancing model effectiveness.

### 6.2 ABLATION ON DATA SIZES

To evaluate the scalability of RANKCLIP, we trained both CLIP and RANKCLIP using 500K, 750K, 1M, and 3M text-image pairs from the CC3M dataset and 15M text-image pairs from the

Table 6: Linear probing top-1 accuracy on 10 downstream datasets. RANKCLIP achieves higher accuracy than CLIP with an average improvement of +5.03% after pretrained on YFCC15M dataset. The results further demonstrate the potential of our approach for applications on large-scale datasets.

| | CIFAR-10 | CIFAR-100 | DTD | FGVGAircraft | Food101 | GTSRB | OxfordPets | SST2 | STL10 | SVHN | Average |
|---|---|---|---|---|---|---|---|---|---|---|---|
| CLIP-15M | 78.72% | 56.46% | 61.70% | 25.44% | 61.65% | 68.69% | 60.78% | 55.24% | 89.95% | 47.98% | 60.66% |
| RANKCLIP-15M | **83.21%** **(+4.49%)** | **62.36%** **(+5.90%)** | **66.06%** **(+4.36%)** | **32.25%** **(+6.81%)** | **68.09%** **(+6.44%)** | **74.14%** **(+5.45%)** | **67.40%** **(+6.62%)** | **56.23%** **(+0.99%)** | **94.15%** **(+4.20%)** | **53.03%** **(+5.05%)** | **65.69%** **(+5.03%)** |

YFCC15M dataset following the same procedure detailed in Appendix A. Fig. 3 *left* presents the zero-shot top-1 classification accuracy on ImageNet1K, where RANKCLIP consistently outperforms CLIP. Notably, it shows a *greater performance increments* as dataset size grows from 1M to 15M pairs, suggesting RANKCLIP's superior scalability, a critical attribute for language-image pretraining. Furthermore, as shown in Table 6, we conducted linear probing on RANKCLIP and CLIP, both pretrained on the 15M text-image pairs, to demonstrate the more promising potential of our method on large-scale datasets.

Fig. 3 *right* illustrates RANKCLIP's robustness across different dataset sizes. The horizontal axis shows the top-1 accuracy on standard ImageNet1K, and the vertical axis on ImageNet1K-R, with a black diagonal line ($y = x$) representing ideal robustness. Any deviation below this line indicates reduced robustness. RANKCLIP consistently stays well above both the red baseline, which reflects typical in-distribution to out-of-distribution generalization (Miller et al., 2021), and close to the ideal line, demonstrating exceptional robustness to distribution shifts.

## 7 ANALYSIS

### 7.1 MODALITY GAP

In this section, we analyze the modality gaps of CLIP and our proposed RANKCLIP by visualizing 250 text-image pair embeddings, reduced to two dimensions using UMAP (McInnes et al., 2018), and complement this with a histogram of the gaps.

Modality gap (Liang et al., 2022) refers to a geometric phenomenon observed in the representation spaces of multimodal models, where different data modalities (like images and texts) are embedded at a noticeable distance from each other, rather than being uniformly distributed as ideally expected. This gap, inherent from initialization and preserved during the contrastive learning process like in CLIP, poses a challenge in language-image pretraining by impacting joint data modeling and understanding. Recent studies (Srivastava & Sharma, 2024; Kumar & Marttinen, 2024; Oh et al., 2024) suggest that reducing this gap could enhance multimodal representations and downstream task performance. The results shown in Fig. 4 indicate that RANKCLIP exhibits a significantly smaller modality gap than CLIP, demonstrating that our ranking consistency approach effectively enhances understanding of text-image semantics.

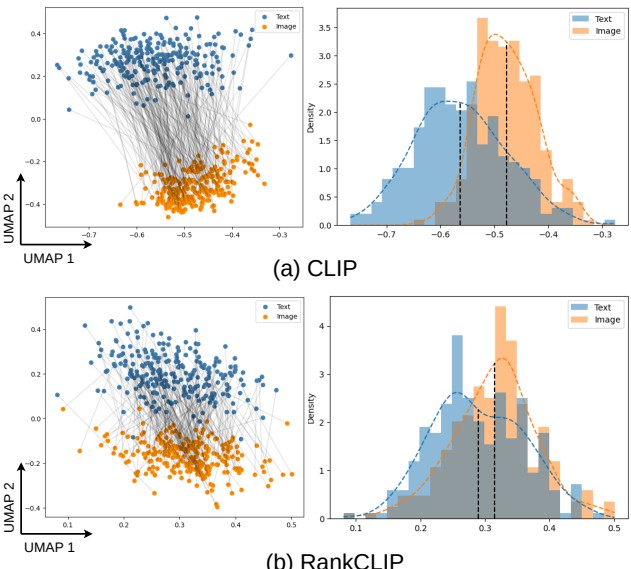

Figure 4: Scatter and histograms plots illustrating modality gaps of (a) CLIP and (b) RANKCLIP.

## 7.2 ALIGNMENT AND UNIFORMITY

Besides alleviating modality gap, it is also commonly believed that a successful contrastive learning method should as well ensure a *broad* and *uniform* distribution covering an hypersphere in space (Wang & Isola, 2020). These two goals, characterized as similarity and uniformity, can be assessed with alignment and uniformity scores, respectively. More specifically, following Goel et al. (2022) and notations defined in §4, we calculate the alignment score $S_A$, and uniformity score $S_U$ to be:

$$S_A = \frac{1}{N} \sum_{j=1}^{N} \hat{I}_j^T \hat{T}_j, \tag{11}$$

$$S_U = \log \left( \frac{1}{N(N-1)} \sum_{j-1}^{N} \sum_{k=1, j \neq k}^{N} \exp^{-\hat{I}_j^T \hat{T}_k} \right) \tag{12}$$

where $N$ is the total number of text-image pairs. Essentially, $S_A$ represents the averaged cosine similarity between text and image embeddings, and $S_U$ averages the dissimilarity measures (exponentiated negative dot products) between all unique pairs of text-image embeddings in the dataset, quantifying how evenly these embeddings are distributed.

A high alignment score represents a strong correlation or similarity between pairs of text-image embeddings, indicating that the images and their textual descriptions are closely aligned in the embedding space. Conversely, a high uniformity score suggests that embeddings are not uniformly distributed; they may be clustering together or not utilizing the embedding space efficiently, which can indicate redundancy in the representations or a lack of diversity. A low uniformity score, on the other hand, suggests that the embeddings are well spread out across the space, indicating a diverse and efficient use of the embedding space, which is generally desirable for tasks like retrieval, where a wide coverage of possible queries are preferred.

As shown in Table 7, we observe that, although CLIP learns representations that are better aligned, as evidenced by its top-ranking alignment scores, these representations fail to achieve uniform distribution across the hypersphere, as highlighted by its significantly higher absolute uniformity scores. On the other hand, RANKCLIP, along with two of its ablated version, RANKCLIP$_I$ and RANKCLIP$_C$, presents much better balance between alignment and uniformity, which results in improved downstream task performance as illustrated in previous experiments as

Table 7: Alignment and uniformity scores of CLIP, RANKCLIP, and its two ablated variants.

| | CIFAR-10 | | | CIFAR-100 | | | ImageNet1K | | |
|---|---|---|---|---|---|---|---|---|---|
| | $S_A$ | $S_U$ | ZS-Top1 | $S_A$ | $S_U$ | ZS-Top1 | $S_A$ | $S_U$ | ZS-Top1 |
| CLIP | **0.40** | -0.35 | 36.35% | **0.42** | -0.35 | 12.22% | **0.44** | -0.29 | 12.08% |
| RANKCLIP | 0.23 | -0.17 | 37.03% | 0.26 | -0.16 | **13.98%** | 0.33 | -0.11 | **17.02%** |
| RANKCLIP$_I$ | 0.24 | -0.16 | **37.47%** | 0.26 | -0.15 | 13.89% | 0.32 | -0.10 | 16.66% |
| RANKCLIP$_C$ | 0.18 | **-0.12** | 28.26% | 0.18 | **-0.10** | 13.29% | 0.26 | **-0.09** | 16.98% |

well as in the representative ZS-Top1 results in Table 7. We also find the results to be informative on a higher level where it indicates that optimizing contrastive learning towards single objective such as alignment or uniformity would not intuitively result in higher downstream task performance.

## 8 CONCLUSION

In this paper, we introduce RANKCLIP, a novel language-image pretraining method that integrates ranking consistency into the contrastive learning paradigm. RANKCLIP aims to better understand the complex many-to-many relationships in diverse text-image pairs by optimizing a self-supervised, list-wise rank loss. Through extensive experiments, including zero-shot classification, robustness to distribution shifts, linear probing, and zero-shot image-text retrieval, RANKCLIP not only enhances performance but also improves model robustness and semantic comprehension, outperforming the baseline CLIP and another state-of-the-art model ALIP. Our ablation studies and analyses further demonstrate and interpret the significance of each component of RANKCLIP in boosting performance and understanding across modalities. We believe that the methodologies and principles of RANKCLIP will inspire further research and lead to the development of models with a deeper understanding of the intricate interactions between visual and textual data.

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

APPENDIX

# A  TRAINING PROCEDURES

## A.1  IMPLEMENTATION DETAILS

For CLIP (Radford et al., 2021), we use the official implementation released by OpenAI[1]. And for ALIP (Yang et al., 2023), we also use the official implementation released by the paper authors[2]. As the proposed RANKCLIP essentially shares the same model architecture (separate vision, text encoders, projection layer, and a classification head) as CLIP, we build upon the CLIP code repository for our model construction[3]. We set the scaling parameters for cross-modal ($\lambda_c$) and in-modal ($\lambda_i$) ranking consistency to 1/16 and 1/16 respectively throughout all the experiments unless otherwise noted. All CLIP, ALIP and RANKCLIP models are initialized from scratch without loading any existing weights. And the embedding sizes for both modalities all project to 1024 across the three models.

## A.2  TRAINING PARAMETERS

Following CLIP (Radford et al., 2021), we adopt the ResNet-50 (He et al., 2016) and transformer architectures (Devlin et al., 2018) for image and text encoding, respectively. Training is conducted from scratch over 64 epochs using a single NVIDIA A100 GPU, with a batch size of 512, an initial learning rate of 0.0005 employing cosine scheduling, and 10,000 warm-up steps.

## A.3  TRAINING TIME CONSUMPTION

we conducted the experiments using the same hardware specifications. The table below shows the time consumption for training our RankCLIP and CLIP models with 50K samples from CC3M using a single NVIDIA A100 GPU.

Table 8: Training Details

|  | Time consumption | Dataset size | epochs | batch_size | model_name |
|---|---|---|---|---|---|
| CLIP | 1d 2h 54m 48s | 50K | 64 | 512 | RN50 |
| RANKCLIP | 1d 1h 4m 23s | 50K | 64 | 512 | RN50 |

As shown in the table, the difference in time consumption is negligible. Interestingly, our method is slightly faster than CLIP, but we think it may be attributed to hardware optimizations or variance.

---

[1]CLIP repository on GitHub: https://github.com/openai/CLIP.

[2]ALIP repository on GitHub: https://github.com/deepglint/ALIP.

[3]RANKCLIP repository will be released upon acceptance.

