# OpenReview forum: "RANKCLIP: Ranking-Consistent Language-Image Pretraining"
_ICLR.cc/2025/Conference — Submitted to ICLR 2025_

### Official Review · Reviewer_ZhYx · 2024-11-01

**Soundness:** 2
**Presentation:** 3
**Contribution:** 2
**Rating:** 5
**Confidence:** 4

**Summary:**

This work introduce RANKCLIP, a novel pretraining method that extends beyond the rigid one-to-one matching framework of CLIP and its variants. By extending the traditional pair-wise loss to list-wise, and leveraging both in-modal and cross-modal ranking consistency, RANKCLIP improves the alignment process, enabling it to capture the nuanced many-to-many relationships between and within each modality.

**Strengths:**

- Overall this paper is well-written and is easy to understand.
- RANKCLIP achieves significant gains in zero-shot classifications over state-of-the-art methods in various downstream tasks.

**Weaknesses:**

- There is a lack of discussion and citation of some related works [A][B], which also propose new alignment objectives for efficient vision-language pre-training. The author should discuss them in the main table results or the related work.

     [A] SaCo Loss: Sample-wise Affinity Consistency for Vision-Language  Pre-training
     [B] ProtoCLIP: Prototypical Contrastive Language Image Pretraining


- The authors demonstrated the effectiveness of the framework on limited image encoder (e.g., ResNet50). In order to verify the generalization ability, the authors should conduct sufficient experimental comparisons on more backbone networks.


- This work resorts to the self-supervised ranking consistency for learning relative semantic similarities. However, without manual labeling, the reference ranking may be noisy and cause the construction of the optimal ranking to be unreliable. As a result, the derived objective does not necessarily learn the relative semantic similarity as the authors mention in the introduction.

**Questions:**

Please see the weaknesses.

---

> ### Author Response · Authors · 2024-11-25
> **Response for Weakness 1**
>
> ### Regarding W1
> >There is a lack of discussion and citation of some related works [A, B], which also propose new alignment objectives for efficient vision-language pre-training. The author should discuss them in the main table results or the related work.
>
> We appreciate the reviewers for pointing out the related works SaCo Loss [A] and ProtoCLIP [B]. Below, we provide a detailed discussion of these works, comparing them to our proposed method, RankCLIP, and highlighting their key differences.
>
> SaCo Loss addresses the Affinity Inconsistency Problem in vision-language pre-training by ensuring that sample-wise affinity relationships within visual and textual modalities remain consistent. The central idea of SaCo Loss is to minimize the disparity between similarity matrices derived from visual and textual embeddings for every pair of samples, thereby improving alignment. However, this approach operates on a pairwise level, focusing on maintaining affinity consistency between individual image-text pairs. In contrast, RankCLIP adopts a list-wise ranking objective, enabling it to optimize both in-modal and cross-modal relationships collectively across entire batches. This approach allows RankCLIP to capture nuanced many-to-many relationships, which SaCo Loss does not explicitly address.
>
> ProtoCLIP enhances representation grouping by introducing prototype-level discrimination. By dynamically constructing and updating prototypes, ProtoCLIP facilitates semantic grouping across modalities and reduces the modality gap through techniques such as Prototypical Back Translation. This method provides stable supervision by leveraging cluster prototypes for alignment and grouping. While ProtoCLIP’s reliance on clustering effectively introduces high-level structural knowledge, it operates on a discrete group level and requires episodic updates of prototypes. RankCLIP, on the other hand, directly optimizes ranking consistency without the need for clustering or explicit prototype construction. Its list-wise optimization framework enables continuous and flexible training, capturing nuanced ranking relationships across all samples in a batch and avoiding the computational overhead associated with clustering.
>
> We will incorporate these discussions of SaCo Loss and ProtoCLIP in the Related Work section of our updated manuscript. We hope we have address the reviewer's comments successfully.
>
> [A] Wu, S., Tan, H., Tian, Z., Chen, Y., Qi, X., & Jia, J. (2024). SaCo Loss: Sample-wise Affinity Consistency for Vision-Language Pre-training. In Proceedings of the IEEE/CVF Conference on Computer Vision and Pattern Recognition (pp. 27358-27369).
> [B] Chen, D., Wu, Z., Liu, F., Yang, Z., Zheng, S., Tan, Y., & Zhou, E. (2023). ProtoCLIP: Prototypical Contrastive Language Image Pretraining. IEEE Transactions on Neural Networks and Learning Systems.

---

> ### Author Response · Authors · 2024-11-25
> **Response for Weakness 2 and 3**
>
> ### Regarding W2
> >The authors demonstrated the effectiveness of the framework on limited image encoder (e.g., ResNet50). In order to verify the generalization ability, the authors should conduct sufficient experimental comparisons on more backbone networks.
>
> We appreciate the reviewer’s suggestion to evaluate our framework using diverse backbone networks to validate its generalization ability. To address this, we conducted zero-shot classification experiments using the **ViT-B/32 backbone** across three benchmarks: CIFAR-10, CIFAR-100, and ImageNet1K. The results are summarized in the table below:
>
> | Method              | CIFAR10 Top1 | CIFAR10 Top3 | CIFAR10 Top5 | CIFAR100 Top1 | CIFAR100 Top3 | CIFAR100 Top5 | ImageNet1K Top1 | ImageNet1K Top3 | ImageNet1K Top5 |
> |---------------------|--------------|--------------|--------------|---------------|---------------|---------------|-----------------|-----------------|-----------------|
> | CLIP-ViT-B/32       | 42.27        | 77.6         | 90.73        | 14.79         | 29.71         | 39.11         | 6.87            | 13.62           | 17.81           |
> | ALIP-ViT-B/32       | 44.03        | 78.17        | 91.18        | 16.10         | **31.84**     | **41.29**     | 7.62            | 14.46           | 18.61           |
> | RankCLIP-ViT-B/32   | **49.15**    | **81.86**    | **93.05**    | **17.43**     | 31.51         | 39.57         | **8.37**        | **15.63**       | **19.72**       |
>
> The experimental results demonstrate that RankCLIP consistently outperforms both CLIP and ALIP across all three benchmarks and evaluation metrics when using the ViT-B/32 backbone. On CIFAR-10, RankCLIP achieves substantial improvements over CLIP in Top-1, Top-3, and Top-5 accuracies, highlighting its ability to extract more refined representations. On CIFAR-100, RankCLIP attains the highest Top-1 accuracy, showcasing its capability to handle complex datasets with fine-grained categories. On ImageNet1K, RankCLIP significantly improves Top-1 accuracy compared to CLIP, further validating its robustness and adaptability to challenging benchmarks.
>
> These results confirm that RankCLIP effectively generalizes to different backbone architectures while maintaining its superior performance. We hope that the additional evaluations address the reviewer's concern and provide compelling evidence of the versatility and robustness of our proposed framework. All the results will be incorporated into the final version of our paper.
>
>
> ### Regarding W3
> >This work resorts to the self-supervised ranking consistency for learning relative semantic similarities. However, without manual labeling, the reference ranking may be noisy and cause the construction of the optimal ranking to be unreliable. As a result, the derived objective does not necessarily learn the relative semantic similarity as the authors mention in the introduction.
>
> We appreciate your observation. To clarify, RankCLIP leverages the natural ranking relationships inherently present in the data, without the need for manual labeling. Specifically, consider an example in the cross-modal setting: ( $image_\text{dog}$, $text_\text{dog}$), ($image_\text{cat}$, $text_\text{cat}$), and ($image_\text{plane}$, $text_\text{plane}$). Our method ensures that the cosine similarity rankings—e.g., $image_\text{dog}$ to $text_\text{cat}$ and $text_\text{plane}$—are consistent with the corresponding rankings for $text_\text{dog}$ to $image_\text{cat}$ and $image_\text{plane}$. This cross-modal alignment effectively integrates the ranking information across modalities.
>
> Moreover, the Plackett-Luce ranking model employed in RankCLIP explicitly minimizes noise by deriving probabilistic rankings from the relative similarity scores of all pairs within a batch. Therefore, our approach ensures robustness, as the model learns to generalize semantic relationships through iterative optimization. By utilizing both in-modal and cross-modal ranking consistencies, RankCLIP enforces a coherent alignment that directly enhances the relative semantic similarity learning process, as substantiated in Section 4.2 and Figure 2 of the manuscript.
>
> We hope this explanation addresses your concern regarding the reliability of the ranking construction and reinforces the foundational strength of our approach.

---

> > ### Comment · Reviewer_ZhYx · 2024-11-25
> >
> > I thank the authors for their comprehensive response. My most concerns have been addressed. However, I still suggest that the authors provide more qualitative example analysis to make it convincing that the model can learn more plausible relative semantic similarities than baseline.

---

### Official Review · Reviewer_BoWp · 2024-11-03

**Soundness:** 3
**Presentation:** 3
**Contribution:** 3
**Rating:** 5
**Confidence:** 4

**Summary:**

This paper introduces **RANKCLIP**, a method for training visual-text embedding models that enforces consistent similarity ranking within and between modalities. The objective comprises two components:

1. **Cross-Modal Rank Consistency**: This ensures that the similarity ranking of text instances to an image sample aligns with the ranking of corresponding images to the text sample.

2. **In-Modal Rank Consistency**: This component maintains that rankings are consistent within each modality, such that text-to-text and image-to-image similarity searches yield comparable ranks.

These two ranking objectives are combined with the traditional CLIP objective to improve the learning of visual-language embeddings.

Experimental results on zero-shot, linear probing, and text-to-image search tasks demonstrate that the proposed objectives enhance embedding quality, yielding more accurate and consistentrepresentations.

**Strengths:**

- **Novelty of Rank Consistency Objective**: The rank consistency objective appears to be novel, as existing methods like the CLIP objective rely on an instance discrimination task focused on distinguishing positive examples from negatives. While embedding-level correlations between similar samples are known to emerge naturally from this approach, no method to date has directly leveraged this correlation. The proposed rank consistency loss effectively utilizes this inherent similarity, potentially improving the overall quality of the learned embeddings.

- **Improvement and Scalability in Embedding Quality**: Consistent improvements in embedding quality are observed across various benchmarks. Notably, the gains become more pronounced as training dataset sizes increase (e.g., from CIFAR to ImageNet to YFCC), indicating promising scalability with larger datasets.

**Weaknesses:**

- **Integration with the Original CLIP Objective**: While the method improves experimental results, further analysis could clarify how the proposed rank consistency objective interacts with the original CLIP objective. For instance, it would be helpful to understand the balance between the two objectives, or if the rank consistency objective alone could effectively learn cross-modal alignment embeddings. This discussion is currently lacking.

- **Limited Ablation Study on Loss Components**: The ablation study on the loss components appears insufficient. Table 5 shows that cross-modal consistency alone performs close to the combined objectives, suggesting that in-modal consistency may have limited impact. This raises questions about whether in-modal consistency is essential, or if the CLIP objective could also benefit from in-modality instance discrimination.

- **Limited comparison with other CLIP modifications**: The paper compares with ALIP and CLIP in experiments. However, there are other recent works on improving the CLIP objective, such as SigLIP,. It is a bit difficult to justify the significance of the quality improvement introduced by the method without comparison with the recent methods and analysis of the comparison results.

Overall, the complementarity between rank consistency and the CLIP objective could be explored further, as it may offer insights beyond the experimental improvements presented. Stopping at the observed gains may overlook a deeper understanding of the method’s theoretical and practical value.

**Questions:**

Please see the weaknesses section for questions regarding the complementarity between rank consistency and the CLIP objective.

Importantly, I would like to see the authors provide a comparison to more baseline methods for advancing CLIP-style cross-modal learning. I am eager to raise my rating if this comparison can be presented.

---

> ### Author Response · Authors · 2024-11-25
> **Response for Weakness 1 and 2**
>
> ### Regarding W1
> >Integration with the Original CLIP Objective: While the method improves experimental results, further analysis could clarify how the proposed rank consistency objective interacts with the original CLIP objective. For instance, it would be helpful to understand the balance between the two objectives, or if the rank consistency objective alone could effectively learn cross-modal alignment embeddings. This discussion is currently lacking.
>
> We appreciate the reviewer’s insightful comment on analyzing the interaction between the proposed rank consistency objective and the original CLIP objective. The rank consistency loss is designed to complement rather than replace the contrastive loss in CLIP. Specifically, our objective enforces global ordering among embeddings while the contrastive loss ensures local pairwise alignment. As detailed in Section 4.4 of the manuscript, rank consistency loss alone cannot effectively learn cross-modal alignments due to its reliance on pre-established pairwise similarities provided by the contrastive framework. Instead, it enhances the embedding space by leveraging unmatched relationships both in-modal and cross-modal, which are typically underutilized in standard CLIP formulations. This synergy ensures improved global coherence while preserving CLIP’s foundational strengths in pairwise alignment.
>
> To clarify the balance, we introduced scaling parameters ($\lambda_1$ and $\lambda_2$ in Equation 10) that modulate the contribution of rank consistency to the overall loss. These parameters ensure that the rank consistency objectives work well with the contrastive loss. We thank the reviewer again for bringing up this meaningful question, and we plan to elaborate further on this point in the revised manuscript.
>
>
> ### Regarding W2
> >Limited Ablation Study on Loss Components: The ablation study on the loss components appears insufficient. Table 5 shows that cross-modal consistency alone performs close to the combined objectives, suggesting that in-modal consistency may have limited impact. This raises questions about whether in-modal consistency is essential, or if the CLIP objective could also benefit from in-modality instance discrimination.
>
> We appreciate the reviewer highlighting the importance of understanding the impact of individual loss components. Our ablation study in Table 5 indeed demonstrates that cross-modal consistency performs well across benchmarks, showing competitive results even compared to the combined objectives on ImageNet. However, it is essential to note that in-modal consistency exhibits superior performance on CIFAR-10 and CIFAR-100, suggesting a dataset-dependent value of this component. This variability underscores that in-modal consistency contributes to capturing intra-modal relationships that might be more critical in smaller or more fine-grained datasets like CIFAR.
>
> Furthermore, while cross-modal consistency addresses semantic alignment between modalities, in-modal consistency enhances structural understanding within each modality, leading to complementary improvements. The combined objectives leverage the strengths of both components to provide balanced and robust performance across diverse datasets. This distinction validates the inclusion of in-modal consistency and supports its utility beyond what is achievable with the CLIP objective alone.
>
> We hope this clarification elucidates the significance of in-modal consistency in our approach and addresses concerns regarding its necessity.

---

> ### Author Response · Authors · 2024-11-25
> **Response for Weakness 3**
>
> ### Regarding W3
> >Limited comparison with other CLIP modifications: The paper compares with ALIP and CLIP in experiments. However, there are other recent works on improving the CLIP objective, such as SigLIP,. It is a bit difficult to justify the significance of the quality improvement introduced by the method without comparison with the recent methods and analysis of the comparison results.
>
> We appreciate the reviewer’s comment regarding the limited comparison with other recent modifications to the CLIP objective. To address this concern, we expanded our experiments to include additional baselines, such as FILIP [1] and SLIP [2]. The results on ImageNet1K, using YFCC15M for pretraining are summarized in table below:
>
> |          | dataset | ImageNet1k |
> | -------- | ------- | ---------- |
> | CLIP     | yfcc15m | 20.63      |
> | FILIP    | yfcc15m | 21.3       |
> | SLIP   | yfcc15m | 28.5       |
> | RankCLIP | yfcc15m | **30.89**  |
>
> These results clearly show that RankCLIP achieves a substantial improvement, outperforming FILIP and SLIP by 9.59% and 2.39%, respectively, and consistently demonstrating superior performance compared to other baseline methods. The significant gains achieved by RankCLIP reflect its ability to exploit many-to-many relationships effectively, enhancing zero-shot classification performance.
>
> We hope this expanded analysis and the inclusion of additional baselines address the reviewer’s concern and illustrate the unique value of our approach.
>
> [1] Yao, L., Huang, R., Hou, L., Lu, G., Niu, M., Xu, H., ... & Xu, C. (2021). Filip: Fine-grained interactive language-image pre-training. arXiv preprint arXiv:2111.07783.
> [2] Mu, N., Kirillov, A., Wagner, D.A., & Xie, S. (2021). SLIP: Self-supervision meets Language-Image Pre-training. In Computer Vision – ECCV 2022: 17th European Conference, Tel Aviv, Israel, October 23–27, 2022, Proceedings, Part XXVI.

---

### Official Review · Reviewer_orHk · 2024-11-03

**Soundness:** 2
**Presentation:** 2
**Contribution:** 2
**Rating:** 3
**Confidence:** 3

**Summary:**

RANKCLIP is a language-image pre-training method that incorporates ranking consistency into contrastive learning to enhance model performance. It seeks to better understand complex many-to-many relationships between diverse text-image pairs by optimizing a self-supervised ranking loss. Extensive experiments show that RANKCLIP improves performance, robustness, and semantic understanding across tasks like zero-shot classification and image-text retrieval, outperforming existing models such as CLIP and ALIP.

**Strengths:**

1.	RANKCLIP is designed to handle the tricky many-to-many relationships between images and text. Instead of just looking at pairs in isolation, it uses a ranking approach to enhance model performance.
2.	When testing against data that’s a little different than what it was trained on, RANKCLIP still holds up well. It also has a knack for understanding semantic nuances, making it better at tasks like image-text retrieval.

**Weaknesses:**

1.	Although it performs well on variants of ImageNet1K with natural distribution shifts, its top-3 and top-5 accuracy on CIFAR-10 is even lower than that of CLIP.
2.	The comparison includes too few SOTA methods; additional methods such as CyCLIP and SoftCLIP should be included to convincingly demonstrate the superiority of the proposed method.

**Questions:**

see weakness.

---

> ### Author Response · Authors · 2024-11-25
> **Response for Weakness 1 and 2**
>
> ## Reviewer 2 -- orHk (Rating: 3)
> ### Regarding W1
> >Although it performs well on variants of ImageNet1K with natural distribution shifts, its top-3 and top-5 accuracy on CIFAR-10 is even lower than that of CLIP.
>
> This is likely because CIFAR-10 with top-3 and top-5 metrics is much simpler, reducing the demand for a deeper model understanding. We conduct an additional experiment to demonstrate our concern. When we pick the earlier epoch(32 epoch) model to evaluate at CIFAR10 and CIFAR100, the top-3 and top-5 performances have outperformed other baseline models.
>
> | Model Type         | CIFAR10 Top1 | CIFAR10 Top3 | CIFAR10 Top5 |
> |--------------------|--------------|--------------|--------------|
> | CLIP              | 36.35        | 70.28        | 85.02        |
> | ALIP              | 35.71        | 72.39        | 88.77        |
> | RankCLIP-27 epoch | **43.11**    | **73.61**    | **89.00**    |
>
>
> ### Regarding W2
> >The comparison includes too few SOTA methods; additional methods such as CyCLIP and SoftCLIP should be included to convincingly demonstrate the superiority of the proposed method.
>
> We appreciate the reviewer’s suggestion to include additional methods. In response, we have incorporated two recent methods, FILIP[1] and SLIP [2], as additional baselines. These methods were pre-trained on the CC15M dataset under the same settings detailed in our draft, and we conducted zero-shot top-1 classification accuracy evaluation on the ImageNet1K dataset. The results are shown in the table below:
>
> |          | dataset | ImageNet1k |
> | -------- | ------- | ---------- |
> | CLIP     | yfcc15m | 20.63      |
> | FILIP    | yfcc15m | 21.3       |
> | SLIP   | yfcc15m | 28.5       |
> | RankCLIP | yfcc15m | **30.89**  |
>
> Regarding the reviewer’s recommendation to include CyCLIP and SoftCLIP, we acknowledge CyCLIP as a notable contribution. However, it is from an earlier publication cycle (NeurIPS 2022), and the baselines we selected provide a more relevant and recent benchmark for comparison. As for SoftCLIP, its implementation and pre-trained weights are not publicly available, which makes it infeasible to reproduce their results accurately for a fair evaluation.
>
> [1] Yao, L., Huang, R., Hou, L., Lu, G., Niu, M., Xu, H., ... & Xu, C. (2021). Filip: Fine-grained interactive language-image pre-training. arXiv preprint arXiv:2111.07783.
> [2] Mu, N., Kirillov, A., Wagner, D.A., & Xie, S. (2021). SLIP: Self-supervision meets Language-Image Pre-training. In Computer Vision – ECCV 2022: 17th European Conference, Tel Aviv, Israel, October 23–27, 2022, Proceedings, Part XXVI.

---

> > ### Comment · Reviewer_orHk · 2024-11-27
> >
> > The authors of the study have included additional comparisons in their experiments.
> > However, they have not provided results comparing CyCLIP and SoftCLIP.
> > This omission cannot show the superiority of rank RANKCLIP.
> > Hence, they do not fully address my concerns.

---

### Official Review · Reviewer_bGF1 · 2024-11-03

**Soundness:** 2
**Presentation:** 2
**Contribution:** 2
**Rating:** 3
**Confidence:** 4

**Summary:**

The paper presents RANKCLIP, a vision-language pretraining method that enhances the alignment between visual and textual modalities. It shifts from traditional pair-wise loss to a list-wise approach, allowing the model to capture many-to-many relationships. RANKCLIP introduces a ranking consistency mechanism to optimize similarity levels within and across modalities. This approach helps the model learn nuanced relationships, such as the closeness between similar images and texts. By leveraging secondary similarities among unmatched pairs, RANKCLIP improves learning efficiency without needing extra data or resources. Comprehensive experiments demostrates the effectiveness of the proposed method in downstream tasks, particularly in zero-shot classification and retrieval accuracy.

**Strengths:**

- Reasonable Motivation: The paper identifies the limitations of existing models like CLIP, discussing the importance of capturing many-to-many relationships in multimodal data, which provides a reasonable motivation for the development of RANKCLIP.

- Extensive Evaluation: RANKCLIP is evaluated across a variety of downstream tasks, including zero-shot image classification, retrieval, and linear probe classification, showing its applicability in different contexts.

**Weaknesses:**

- Lack of Discussion on Related Works: The paper does not adequately discuss other works that also aim to construct many-to-many relationships in vision-language pretraining. For example, [1] proposed a progressive self-distillation method that uses image-to-text logits (and vice versa) as targets, while [2] introduced in-modal consistency.

- Lack of Novelty: RANKCLIP closely resembles the method described in [1], raising questions about its novelty.

- Misaligned Experiment Settings: The experimental setup is misaligned, making the results less convincing. While many CLIP-related works utilize the ViT-B/32 architecture as the vision backbone, RANKCLIP employs RN50, which could affect the comparability of the results.

- Performance Downgrade in Linear Probe Classification: The proposed method underperforms in linear probe classification on fine-grained datasets, such as GVGAircraft, Food101, and GTSRB. The paper does not address this phenomenon, which limits the interpretation of its effectiveness.

- Unconvincing Results in Zero-Shot Text/Image Retrieval: There is a substantial disparity between the results of image retrieval and text retrieval (84.1% vs. 8.1%), which raises doubts about the reliability of these findings.


[1] Andonian, Alex, Shixing Chen, and Raffay Hamid. "Robust cross-modal representation learning with progressive self-distillation." Proceedings of the IEEE/CVF Conference on Computer Vision and Pattern Recognition. 2022.

[2] Goel, Shashank, et al. "Cyclip: Cyclic contrastive language-image pretraining." Advances in Neural Information Processing Systems 35 (2022): 6704-6719.

**Questions:**

Please refer to the weaknesses.

---

> ### Author Response · Authors · 2024-11-25
> **Response for Weakness 1 to 3**
>
> ### Regarding W1 and W2
> >W1: Lack of Discussion on Related Works: The paper does not adequately discuss other works that also aim to construct many-to-many relationships in vision-language pretraining. For example, [1] proposed a progressive self-distillation method that uses image-to-text logits (and vice versa) as targets, while [2] introduced in-modal consistency.
> >W2: Lack of Novelty: RANKCLIP closely resembles the method described in [1], raising questions about its novelty.
>
>
> We deeply appreciate the reviewer for highlighting these important works and providing thoughtful observations regarding their relevance to RankCLIP. In fact, CyCLIP [2] inspired our work and we have discussed it in line 88 of the original draft when introducing our work.
>
> In terms of [1], we agree that the proposed progressive self-distillation framework is indeed noteworthy. It introduces a dynamic mechanism where the model partitions aligned and unaligned instances, generating soft-alignment targets for subsets of image-text pairs. Specifically, it incorporates a weighted cross-entropy loss for image-to-text (v2t) and text-to-image (t2v) alignments, computed using selected instances and teacher-provided soft targets.
>
> That said, we would like to clarify a key distinction between RankCLIP and the mentioned frameworks. Both [1] and [2] are fundamentally pair-wise algorithms, focusing on one-to-one relationships. In contrast, RANKCLIP adopts a list-wise paradigm, which represents a significant conceptual shift. Our motivation stems from recognizing that existing methods fail to exploit the rich relative rank information present in datasets. This information is essential for learning consistent and robust representations across modalities. By leveraging list-wise batch ranking relationships rather than pair-wise alignments, RankCLIP fundamentally redefines how many-to-many relationships are modeled in vision-language pretraining.
>
> We thank the reviewer again for their insightful feedback, which has helped us better articulate RankCLIP’s unique contributions. We will ensure that this distinction is explicitly discussed in the final version of our paper.
>
>
> ### Regarding W3
> >Misaligned Experiment Settings: The experimental setup is misaligned, making the results less convincing. While many CLIP-related works utilize the ViT-B/32 architecture as the vision backbone, RANKCLIP employs RN50, which could affect the comparability of the results.
>
> We appreciate your feedback regarding the experimental setup. To address your concern and ensure fair comparability, we conducted additional experiments using the ViT-B/32 as the vision backbone. These experiments evaluate zero-shot top-1, top-3, and top-5 classification accuracy on CIFAR-10, CIFAR-100, and ImageNet1K. The results are shown in the table below:
>
> | Method              | CIFAR10 Top1 | CIFAR10 Top3 | CIFAR10 Top5 | CIFAR100 Top1 | CIFAR100 Top3 | CIFAR100 Top5 | ImageNet1K Top1 | ImageNet1K Top3 | ImageNet1K Top5 |
> |---------------------|--------------|--------------|--------------|---------------|---------------|---------------|-----------------|-----------------|-----------------|
> | CLIP-ViT-B/32       | 42.27        | 77.6         | 90.73        | 14.79         | 29.71         | 39.11         | 6.87            | 13.62           | 17.81           |
> | ALIP-ViT-B/32       | 44.03        | 78.17        | 91.18        | 16.10         | **31.84**     | **41.29**     | 7.62            | 14.46           | 18.61           |
> | RankCLIP-ViT-B/32   | **49.15**    | **81.86**    | **93.05**    | **17.43**     | 31.51         | 39.57         | **8.37**        | **15.63**       | **19.72**       |
>
> The results further demonstrate the effectiveness of RankCLIP across datasets and evaluation metrics. For CIFAR-10, RankCLIP achieves the best performance, significantly surpassing CLIP and ALIP in top-1, top-3, and top-5 accuracy. On CIFAR-100, RankCLIP achieves the highest top-1 accuracy of 17.43%, outperforming CLIP and ALIP by 2.64% and 1.33%, respectively. Although ALIP slightly outperforms RankCLIP in top-3 and top-5 metrics on CIFAR-100, RankCLIP’s superior top-1 performance highlights its strength in precise zero-shot predictions. On the more challenging ImageNet1K dataset, RankCLIP achieves consistent improvements over both baselines across all metrics. It outperforms ALIP in top-1 accuracy by 0.75% and CLIP by 1.50%.
>
> Overall, these results align consistently with the original results utilizing the RN50 backbone, validating the strength of RankCLIP in enhancing language-image pretraining performance.

---

> ### Author Response · Authors · 2024-11-25
> **Response for Weakness 4 and 5**
>
> ### Regarding W4
> > Performance Downgrade in Linear Probe Classification: The proposed method underperforms in linear probe classification on fine-grained datasets, such as GVGAircraft, Food101, and GTSRB. The paper does not address this phenomenon, which limits the interpretation of its effectiveness.
>
> We appreciate the reviewer’s observation regarding the linear probe classification performance on datasets such as GVGAircraft, Food101, and GTSRB. First, we would like to point out that our method demonstrates a **higher average accuracy** compared to the baseline methods across all the evaluated datasets, as shown in Table 3 of the draft, which indicates that RankCLIP achieves consistent improvements in broader settings. Second, performance variation across datasets is not uncommon and has been observed in prior works as well. For instance, both ALIP and CyCLIP [2] exhibit significant degradation on certain datasets while excelling on others.
>
> ### Regarding W5
> >Unconvincing Results in Zero-Shot Text/Image Retrieval: There is a substantial disparity between the results of image retrieval and text retrieval (84.1% vs. 8.1%), which raises doubts about the reliability of these findings.
>
> We acknowledge the observed disparity between zero-shot image and text retrieval performances. This difference primarily arises due to the inherent asymmetry in the retrieval task dynamics: while image retrieval benefits from well-structured visual embeddings aligning to textual descriptions, text retrieval demands identifying textual similarities to visual details, often involving finer-grained semantic understanding across resolutions and object contexts. The complexity of such cross-modal associations, particularly in datasets requiring nuanced reasoning, contributes to the observed variance. Additionally, similar performance gaps have been reported in prior works, such as CyCLIP [2], which aligns with our findings. We hope this clarifies the rationale behind the disparity and provides greater confidence in the reliability and significance of our experimental results.

---

> > ### Comment · Reviewer_bGF1 · 2024-11-26
> >
> > Thanks to the authors for the rebuttal. However, I don't agree with the view that the rebuttal claims [1] and [2] are fundamentally pairwise algorithms, focusing on one-to-one relationships. [1] and [2] were proposed to formulate many-to-many relationships. Additionally, I believe the paper avoids mentioning CyCLIP, but the rebuttal states that CyCLIP inspired the proposed work, which is not explicitly indicated in the paper. Therefore, I would like to maintain my score.

---

### Meta-Review · Area_Chair_nFhJ · 2024-12-19

**Metareview:**

This paper introduces RANKCLIP, a method designed to capture many-to-many relationships in both in-modal and cross-modal comparisons. It addresses the limitations of existing contrastive learning methods, such as CLIP, which rely solely on pairwise matching. Experiments demonstrate improved performance compared to baseline methods like CLIP, ALIP, and FLIP.

However, as noted by the reviewers, the approach shares similarities with CyCLIP and ProtoCLIP, raising concerns about its novelty. Additionally, the paper lacks direct comparisons to these methods. Furthermore, the reliance on self-supervised ranking consistency without manual labeling may introduce noise, potentially undermining the effectiveness of the semantic similarity learning process.

Four reviewers provided negative feedback on this work, and the AC concurs with their assessment. The method holds potential, and with more comprehensive comparisons, ablations, and clarity on contributions, it could make a stronger impact in future submissions.

**Additional Comments On Reviewer Discussion:**

The authors have actively addressed the reviewers’ concerns, and the inclusion of additional comparisons along with clarifications of the proposed contributions offered some reassurance regarding the method’s validity. However, the absence of direct comparisons with CyCLIP and ProtoCLIP, insufficient ablation studies, and the limited evaluation scope leave notable gaps in establishing the novelty and robustness of the approach.

---

### Decision · Program_Chairs · 2025-01-22

Reject